

# Nanotechnology in action: silver nanoparticles for improved eco-friendly remediation

Suresh Babu Naidu Krishna[1,2], Abdul Gaffar Sheik[1], Karen Pillay[2], Manhal Ahmed Hamza[3], Mohammed Yagoub Mohammed Elamir[4] and Samy Selim[4]

[1] Institute for Water and Wastewater Technology, Durban University of Technology, Durban, KwaZulu-Natal, South Africa
[2] Department of Biochemistry, School of Life Sciences, University of KwaZulu-Natal, Durban, KwaZulu-Natal, South Africa
[3] Department of Medical Microbiology, Faculty of Medical Laboratory Sciences, Omdurman Islamic University, Omdurman, Sudan
[4] Department of Clinical Laboratory Sciences, College of Applied Medical Sciences, Jouf University, Sakaka, Saudi Arabia

Corresponding author
Samy Selim, sabdulsalam@ju.edu.sa

## ABSTRACT

Nanotechnology is an exciting area with great potential for use in biotechnology due to the far-reaching effects of nanoscale materials and their size-dependent characteristics. Silver and other metal nanoparticles have attracted a lot of attention lately because of the exceptional optical, electrical, and antimicrobial characteristics they possess. Silver nanoparticles (AgNPs) stand out due to their cost-effectiveness and abundant presence in the earth's crust, making them a compelling subject for further exploration. The vital efficacy of silver nanoparticles in addressing environmental concerns is emphasized in this thorough overview that dives into their significance in environmental remediation. Leveraging the distinctive properties of AgNPs, such as their antibacterial and catalytic characteristics, innovative solutions for efficient treatment of pollutants are being developed. The review critically examines the transformative potential of silver nanoparticles, exploring their various applications and promising achievements in enhancing environmental remediation techniques. As environmental defenders, this study advocates for intensified investigation and application of silver nanoparticles. Furthermore, this review aims to assist future investigators in developing more cost-effective and efficient innovations involving AgNPs carrying nanoprobes. These nanoprobes have the potential to detect numerous groups of contaminants simultaneously, with a low limit of detection (LOD) and reliable reproducibility. The goal is to utilize these innovations for environmental remediation purposes.

## INTRODUCTION

Environmental pollution is a major global dilemma that is progressively worsening and causing serious and irreversible damage to the planet (*Das, Sen & Debnath, 2015*; *Ortúzar et al., 2022*). Urbanization and the ever-growing human population have pushed the limits of resource use, which eventually contributes to the deterioration of nature. Carbon monoxide, chlorofluorocarbons, volatile organic compounds, hydrocarbons, and nitrogen oxides are only few of the many pollutants that are found in the air today (*Feizi et al., 2023*; *Rani et al., 2023*; *Turdimovich & Khasanovich, 2023*). Arsenic, heavy metals, and chlorinated chemicals pollute water and soil with some of the major sources of pollution being sewage, industrial effluents, indiscriminate use of pesticides and fertilizers, and oil spills (*Elbadawy et al., 2023*; *Usman et al., 2020*).

Nanomaterials are materials with unique qualities at the nanoscale level that have attracted a lot of interest because of their unique physical, chemical, and mechanical capabilities (*El-Saadony et al., 2023*; *Ngcongco, Krishna & Pillay, 2023*). The increased surface area-to-volume ratio of nanoparticles (NPs) improves their reactivity and adsorption capability (*Kumari, Alam & Siddiqi, 2019*; *Shankara et al., 2022*). This property thus enables NPs to effectively remove pollutants and contaminants such as heavy metals, organic compounds, and even pathogens from wastewater. Iron oxide NPs have been shown to be effective in the removal of heavy metals *via* adsorption and coagulation processes (*Jabbar, Barzinjy & Hamad, 2022*), and carbon-based nanomaterials, such as activated carbon nanotubes and graphene oxide, have high adsorption capacities, making them useful for removing organic contaminants and colourants (*Amil Usmani et al., 2017*; *Joy et al., 2023*; *Kosgey et al., 2022*). Furthermore, metal oxide NPs such as titanium dioxide and zinc oxide have photocatalytic characteristics that allow the breakdown of organic contaminants when exposed to ultraviolet (UV) light (*Pal et al., 2022*; *Prakash et al., 2022*). Nanomaterials are thus a promising new technology for environmental remediation, and further research is needed to explore their full potential.

AgNPs are highly distinctive and appealing due to their low cost of manufacturing, environmental sustainability, and, most outstandingly, their high toxicity primarily to multidrug-resistant microorganisms whilst being non-cytotoxic to healthy cells at low dosages (*Nie, Zhao & Xu, 2023*). As a result, it now has multifaceted application (*Guerra et al., 2018*; *Mo, Zhou & He, 2022a*; *Nakamura et al., 2019*; *Ngcongco, Krishna & Pillay, 2023*), having potential to be effective antibacterial agents as well as showing promise for their use in environmental remediation processes (*Tarrat & Loffreda, 2023*). AgNPs-based nanomaterials with their antibacterial, optical, and electrical properties are thus at the forefront of nanotechnology, with applications in environmental disinfection, pollutant elimination, environmental monitoring, and energy conversion (*Del Prado-Audelo et al., 2021*; *Durgalakshmi, Rajendran & Naushad, 2019*; *Gracia-Pinilla et al., 2008*; *Khin et al., 2012*; *Mo, Zhou & He, 2022b*; *Ningthoujam et al., 2022*). Considering the safe and biocompatible nature of the AgNPs, it becomes worthy to apply it to environment remediation purposes in the form of electro sensors, photocatalysts, and fluorogenic probes (*Abraham et al., 2013*; *Bourgonje et al., 2023*).

This review highlights advanced eco-friendly remediation applications and fundamental principles to guide future research on AgNPs and their integration into functional materials. Furthermore, it discusses silver nanoparticle application as electro sensors, photocatalysts, and fluorogenic probes.

## METHODOLOGY

An extensive review of the literature encompassed studies from esteemed electronic databases, such as Scopus, Google Scholar, MEDLINE, and ScienceDirect. Special emphasis was placed on the biological and chemical synthesis of AgNPs. The search was conducted using various keyword combinations relevant to the subject, including "biosynthesis silver nanoparticles" OR 'silver nanoparticles' OR 'biosynthesis' OR 'green synthesis' OR "silver based nanocomplexes" OR "environmental remediation" AND ('antibacterial activity' OR 'antimicrobial resistance'). Most of these publications are in fields such as chemistry, materials science, physics, engineering, polymer science, spectroscopy, electrochemistry, molecular biochemistry, optics, and spectroscopy, spanning from 2007 to 2024 (*Tran, Nguyen & Le, 2013*). Specific inclusion and exclusion criteria were established to determine which studies should be considered, considering factors such as the research methodology, publication year, and primary research focus. The titles and abstracts of the identified studies were scrutinized to assess their alignment with inclusion criteria. Studies failing to meet these criteria were eliminated at this stage. Relevant data was extracted from the included studies (*Magdy et al., 2024*).

### Biosynthesis of AgNPs
#### Conventional strategies vs. biological synthetic strategies
Metallic silver is a soft, white, shiny and rare element that is naturally available and has good thermal and electrical conductivity (*Islam, Jacob & Antunes, 2021*; *Nie, Zhao & Xu, 2023*). AgNPs are a type of metallic silver that is less than 100 nm in at least one dimension, enabling the NPs to have a high surface area to volume ratio (*Antunes et al., 2017*). Current methodologies for AgNP synthesis and other metal preparations can be divided into two categories, namely the "top to bottom" approach which is typically used by physicists, and the "bottom to up" approach which is commonly used by chemists (*Mo, Zhou & He, 2022b*; *Moodley et al., 2018*). Both approaches converge on the nano dimension, but their synthetic technologies are very different from one another. In "top to bottom" procedures, several physical methods are used to break down bulk solid materials into their nanoparticulate form. These physical methods include grinding, milling, sputtering, evaporation-condensation, and thermal/laser ablation. "Bottom to up" procedures involve a variety of chemical and biological processes to produce NPs *via* the self-assembly of atoms such as $Ag^+$ into nuclei, which then evolve into nano-sized particles.

Two of the most popular techniques for the "top to bottom" approach are evaporation-condensation and laser ablation (*Moodley et al., 2020*). Evaporation-condensation is accomplished using a tube furnace operating at room temperature, with the primary material (metal Ag) contained in a boat that is positioned in the middle of the

furnace and vaporized into a carrier gas (*Ahmed et al., 2016*). This method has been shown to have a few flaws, such as the fact that the furnace takes up a considerable amount of space, it necessitates a significant amount of energy to raise the temperature of the atmosphere surrounding the source material, and also necessitates a significant amount of time to achieve thermal stability. However, one of the most significant drawbacks of this method of synthesis is that it produces defects in the surface structure of the produced NPs, which ultimately can change the physical characteristics of the NPs (*Iravani et al., 2014*; *Sportelli et al., 2018*).

Irradiation is utilized in the process of laser ablation to strip material away from a bulk metal that is in solution. The efficacy of this method and the features of the nascent particles are substantially determined by a variety of parameters. These parameters include the wavelength of the laser, the duration of laser pulses, the laser fluence, the ablation time, and the effective liquid medium with or without surfactants (*Chen & Yeh, 2002*; *Kim et al., 2005*). The elimination of contaminants in solution that have the potential to contaminate the nanoparticle preparation is a significant benefit that can be gained from using laser ablation for the preparation of AgNP (*Tsuji et al., 2002*).

With regard to the "bottom to up" approaches, wet chemical reduction is the most frequently used method for the synthesis of NPs (*Iravani et al., 2014*), despite the fact that various other ways have been documented (*Amin, Pazouki & Hosseinnia, 2009*; *Yang & Pan, 2012*). Wet chemical reduction involves the reduction of a metal salt precursor in an organic or aqueous solution. Ascorbate, borohydride, citrate, elemental hydrogen, formaldehyde, N-N-dimethyl formamide (DMF), Tollen's reagent, and polyethylene glycol blocks are some of the organic and inorganic chemicals that have been successfully utilized as reducing agents in the production of AgNPs (*Bagheri et al., 2023*; *Iravani, 2011*; *Radwan et al., 2021*; *Thangavelu et al., 2022*). In order to stop the formation of aggregates of newly formed NPs, the reaction solution contains, in addition to reducing agents, protective stabilizing agents (*Bai et al., 2007*; *Kapoor et al., 1994*). Once stability is reached, this approach has the potential to be effective in producing high yields of NPs while maintaining low costs of preparation (*Song et al., 2009*). Nevertheless, the effectiveness of this technology is called into question due to the possibility of nascent NPs being contaminated by precursor chemicals, the utilization of hazardous solvents, and the production of toxic by-products (*Dhuper, Panda & Nayak, 2012*; *Iravani et al., 2014*; *Thakkar, Mhatre & Parikh, 2010*).

The physical and chemical approaches outlined above have limitations that hinder their implementation in the synthesis of NPs for biological application (*Islam, Jacob & Antunes, 2021*). In this context, significant endeavors have been made to formulate nanoparticle synthesis methods that adhere to environmentally sustainable principles. In essence, this would involve the utilization of harmless biotechnological instruments and has led to the emergence of the notion of environmentally friendly or green technology. This method can be most accurately characterized as the utilization of biological pathways, specifically involving plants, microbes, or their by-products, in the process of synthesizing NPs (*Patra & Baek, 2015*; *Singh et al., 2019*). The bio-inspired technologies depicted in Fig. 1 exhibit

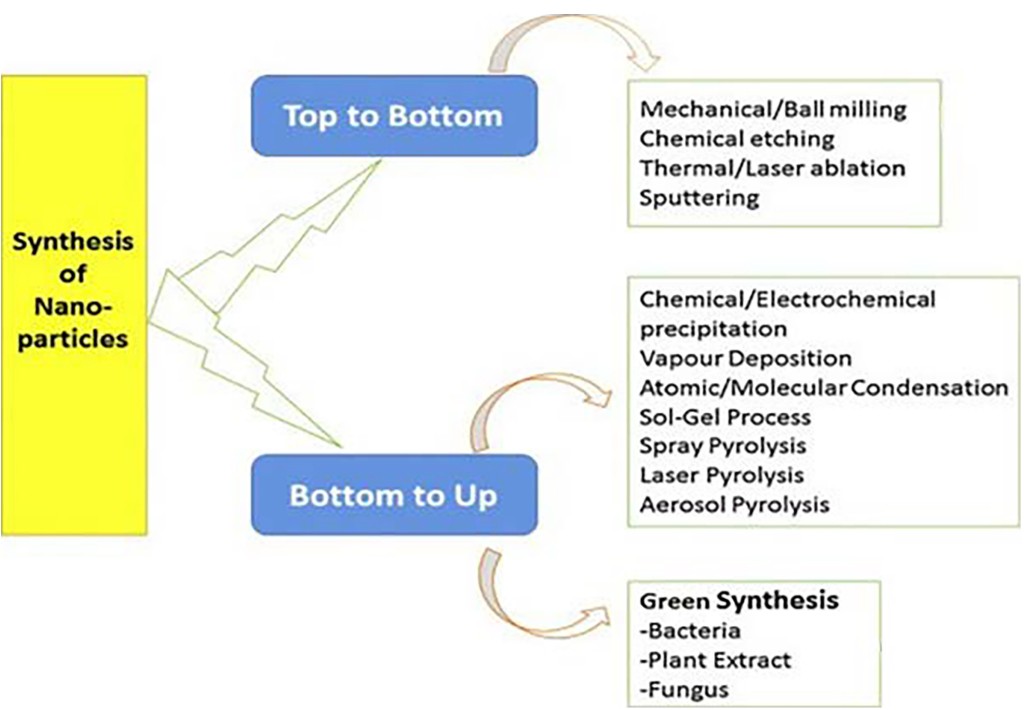

**Figure 1** Multiple techniques of nanoparticle synthesis (*Mtambo, Krishna & Sershen, 2019*).

not only environmental friendliness, but also cost-effectiveness and the potential for seamless expansion to accommodate large-scale manufacturing (*Ahmed et al., 2016*).

As highlighted earlier, biological methods for the synthesis of AgNP involve the utilization of living organisms or the extracts of such organisms as capping or reducing agents in a synthetic reaction. To date, numerous types of biological entities, such as viruses (*Velusamy et al., 2016*), bacteria (*Thakkar, Mhatre & Parikh, 2010*), plants (*Iravani & Varma, 2019*), algae (*Khanna, Kaur & Goyal, 2019*), fungi (*Joy et al., 2023*), yeast (*Skalickova, Baron & Sochor, 2017*), and mammalian cells (*Nazarenus et al., 2014*), have been investigated to determine their capability to produce AgNPs. The process underlying biological synthesis can be broken down into two distinct phases: the bio reduction phase and the biosorption phase. Bio reduction takes place when metal ions go through the process of chemical reduction and become complexes that are biologically stable. A wide variety of organisms have been seen to exhibit dissimilatory metal reduction, which involves the coupling of enzyme reduction with enzyme oxidation. The resultant NPs can be safely removed from the reaction mixture when they have become stable and inert. Alternately, biosorption refers to the process of attaching metal ions onto a living entity, such as the cell wall. There are several species of bacteria, fungi, and plants, each of which can produce peptides or possess modified cell wall structures that are capable of binding metal ions and, as a result, producing stable complexes in the shape of NPs (*Pantidos & Horsfall, 2014*). To obtain a comprehensive understanding of the literature, we suggest readers consult scholarly review articles and book chapters that have been published within the past 2 years (*Bagheri et al., 2023*; *Bourgonje et al., 2023*; *Bruna et al., 2021*;
*Husain et al., 2023*; *Jaswal & Gupta, 2023*; *Thangavelu et al., 2022*; *Yaqoob, Umar & Ibrahim, 2020*).

## Application of AgNPs for environmental remediation

### *AgNPs as sensors*

The amalgamation of nanotechnology with sensor technology has resulted in a synergy that has revolutionized environmental monitoring. By integrating AgNPs into sensors, scientists and engineers have unlocked incredible opportunities to detect and analyze environmental pollutants (Table 1) with unprecedented precision (*Li, Cushing & Wu, 2015*). Many studies have shown that AgNPs are perfect for use as sensors. Preferably, particles of lower size will enable reactant molecules to access more silver atoms, making them a potential electrochemical sensor (*Ivanišević, 2023*). However, AgNPs with a zero-net-charge tend to cluster to form larger aggregates with lower surface area to volume ratios. In recent years, significant progress has been achieved in synthesizing AgNPs with regulated shape, size, surface charge, and physicochemical properties to preserve their distinctive thermal and electrical capabilities. AgNPs are often amalgamated with another metallic material (*Manivannan et al., 2018*), embedded into single- (*Bezerra et al., 2021*) or multiwalled (*Wan et al., 2017*) carbon nanotubes, anchored on functionalized graphene (oxide) platforms (*Bao et al., 2021*; *Cheng et al., 2019*), modified to form multifunctional ternary systems (*Abraham et al., 2020*), or deposited as a thin film on the electrode surface (*Meng et al., 2018*) to improve their sensing properties.

In a recent study, graphite carbon sheets were utilized to create AgNPs, which are then employed as an electrochemical sensor to detect organic compounds in water samples (*Ivanišević, 2023*). Extensive analysis was conducted on the developed nanomaterial using scanning electron microscopy (SEM), X-ray diffraction technique (XRD), and IR spectroscopy to examine its structure, composition, and morphology (*Zahran et al., 2021*). The nitrofurazone sensor demonstrated remarkable sensitivity in detecting the substance, with $1.2 \times 10^{-8}$ M for the limit of detection and $1.3 \times 10^{-7}$ M for the limit of quantification. The sensor showed a reduction peak at −0.57V, and a calibration curve was created using concentrations ranging from $10^{-4}$ to $2 \times 10^{-7}$ M, indicating the reduction of nitrofurazone (*Zoubir et al., 2022*). Through experimental verification, it has been shown that the use of AgNPs on the carbon nanosheets significantly improves their electrocatalytic ability and enhances their potential for reducing nitrofurazone (*Zoubir et al., 2022*). The sensor performed excellently and is suitable for detecting nitrofurazone in various aqueous samples, including water from the tap, human faeces, and commercial milk. Furthermore, it demonstrated exceptional consistency and the ability to be used repeatedly (*Zoubir et al., 2022*).

A new study has introduced a highly accurate and efficient colorimetric technique for identifying chromium (III) in tap water (*Qadri et al., 2022*). This method utilizes AgNPs that have been functionalized with a derivative of phenyl benzotriazole (PBT-AgNPs) (*Qadri et al., 2022*). At room temperature, the PBT-functionalized NPs were produced through a reduction process using sodium borohydride (NaBH4). Multiple approaches were used to investigate the NPs, including UV–Vis spectroscopy, Zetasizer, Fourier

**Table 1 List of silver-based nano-complexes that have shown potential as detection agents of environmental pollutants in recent years (2016–2023).**

| Nano-complex | Molecule detected | Application | Efficiency expressed as limit of detection (LOD) | Reference |
|---|---|---|---|---|
| Silver nanoparticles synthesized on graphite carbon sheets | Nitrofurazone | A sensor for the detection of nitrofurazone, a possible teratogen and carcinogen, in aqueous systems. | $1.2 \times 10^{-8}$ M | Zoubir et al. (2022) |
| Phenylbenzotriazole (PBT) derivative functionalized silver nanoparticles (PBT-AgNPs). | Cr (III) | A sensing probe for detection of Cr (III) during monitoring of water systems, and which has excellent selectivity even in the presence of other interfering metals ions. | 0.2 μM | Qadri et al. (2022) |
| Curcumin functionalized silver nanoparticles | Paracetamol | To monitor the discharge of pharmaceutical pollutant in water effluent since paracetamol in high dosage can cause organ damage | 0.29 μM | Kumar et al. (2022) |
| Graphene oxide (rGO)-wrapped dual-layer silver nanoparticles (AgNPs) on titania nanotube (TiO$_2$ NTs) arrays | Glyphosate | To monitor glyphosate levels using a surface-enhanced Raman scattering (SERS) substrate. Glyphosate is a widely used organophosphate herbicide in agricultural applications that contaminates the environment. | 3 μg/L, which is below the maximum contaminant level of glyphosate in environmental water, as recommended by the U.S. EPA and the European Union. | Butmee, Samphao & Tumcharern (2022) |
| AgNPs functionalized with mercaptoundecanoic acid (11-MUA). | Nickel ions | Surface plasmon resonance (SPR) colorimetric sensor of toxic nickel ions especially from industrial water effluent. | Micromolar levels with acceptable selectivity in the presence of $Mn^{2+}$, $Co^{2+}$, $Cd^{2+}$, $Cu^{2+}$, $Zn^{2+}$, $Fe^{2+}$, $Hg^{2+}$, $Pb^{2+}$, and $Cr^{3+}$. | Rossi et al. (2021) |
| Gold-silver nanoparticles | Cr (VI) and Cr (III) | An electrochemical sensor array for detection of Cr (VI) and Cr (III) in wastewater treatment processes. | 0.1 ppb for both Cr (VI) and Cr (III) | Zhao et al. (2021) |
| Silver nanoparticles (AgNP) with different capping agents | Acidic and oxidizing gases and other air pollutants | A colorimetric sensor array for quantitative identification of 11 common pollutants relevant to the protection of cultural heritage objects and for museum environmental monitoring. | Sub-ppb for 1 h exposures | Li et al. (2020) |
| Silver nanoparticles synthesized using *Ginkgo biloba* | Chromium (VI) | A fluorescent probe for detection of the toxic hexavalent chromium, especially in water systems. | 0.014 μM | Huang et al. (2020) |
| Silver-poly (methyl methacrylate) nanoparticles | Hydrogen peroxide | A colorimetric sensor for the detection of hydrogen peroxide, a toxic contaminant that can be found in food, pharmaceuticals, and environmental processes. | $10^{-6}$ M. | Carbone et al. (2019) |
| Silver nanoparticles synthesized from *Agaricus bisporus* | Hg (II) | Detection of Hg (II) ions without the use of modifiers or sophisticated instrumentation. The Hg (II) ion is very toxic, however due to its stable form, it is highly soluble in water thus leading to severe environmental concerns. | $2.1 \times 10^{-6}$ M. | Sebastian, Aravind & Mathew (2018) |

(Continued)

| Table 1 (continued) | | | | |
|---|---|---|---|---|
| Nano-complex | Molecule detected | Application | Efficiency expressed as limit of detection (LOD) | Reference |
| Silver nanoparticle–reduced graphene oxide–polyaniline (AgNPs–rGO–PANI) nanocomposite | Hydrogen peroxide | A sensor for the detection of hydrogen peroxide, a toxic contaminant that can be found in food, pharmaceuticals, and environmental processes. | 50 nM | *Kumar et al. (2018)* |
| Gold nanostar (Au NS) core–silver nanoparticle (Ag NP) satellites | Aflatoxin B1 | Surface-enhanced Raman scattering (SERS) sensor for detection of Aflatoxin B1 in the environment, especially in raw food materials, such as grains, corn, feedstuffs and peanuts. Aflatoxins are toxic and recognized as potent carcinogens, mutagens, and teratogens. | 0.48 pg/mL | *Li et al. (2016)* |

Transformer InfraRed (FTIR), scanning electron microscope (SEM), and atomic force microscopy (AFM). Despite the variations in temperature, pH, and ionic strength of the electrolytes, the stability of the NPs remained largely unaffected. After conducting tests, it was discovered that the PBT-AgNPs functioned as a probe with remarkable sensitivity to the presence of Cr (III) (*Qadri et al., 2022*). Additionally, they exhibited exceptional selectivity when other interfering metal ions were present (*Qadri et al., 2022*). The Jobs plot revealed a binding ratio of 1:2 between the NPs and Cr (III). The quenching of the surface plasmon resonance (SPR) band, as the concentration varies, exhibits a highly linear response ($R^2$ = 0.9992), with a remarkable limit of detection of 0.2 µM. Additionally, the PBT-AgNPs were utilized as a sensing probe to detect the presence of Cr (III) in tap water samples (*Qadri et al., 2022*). These experimental findings demonstrate the use of supramolecular stabilized AgNPs as a straightforward, precise, and convenient alternative for detecting Cr (III).

Novel sensors were developed by *Zhao et al. (2021)* which were modified with nano silver-gold and silver-gold oxides (Fig. 2). For this study, a new electrochemical sensor array was created using modified electrodes with gold-AgNPs. The goal was to detect both chromium species (Cr (III) and (VI)) simultaneously. In this study, the screen-printed carbon electrodes (SPCEs) were enhanced with silver-gold bimetallic NPs using electrochemical deposition. For the detection of Cr (III), the silver-gold bimetallic NPs were oxidized to create stable silver-gold bimetallic oxide NPs. Based on the findings, it was observed that incorporating silver, at a theoretical value of 1% of gold, had a positive impact on the creation and maintenance of oxides on the surface of gold NPs. After conducting thorough characterization, the two types of electrodes were combined to create an electrochemical sensor array capable of detecting Cr (VI) and Cr (III) with high selectivity and sensitivity. For Cr (VI), the linear range and limit of detection (LOD) were determined to be 0.05–5 ppm and 0.1 ppb, respectively, based on a three times signal-to-noise ratio. As for Cr (III), the linear range and LOD were found to be 0.05–1 ppm and 0.1 ppb, respectively. After extensive testing, the electrochemical sensor array successfully detected Cr (VI) and Cr (III) in various samples such as tap water, artificial saliva, and

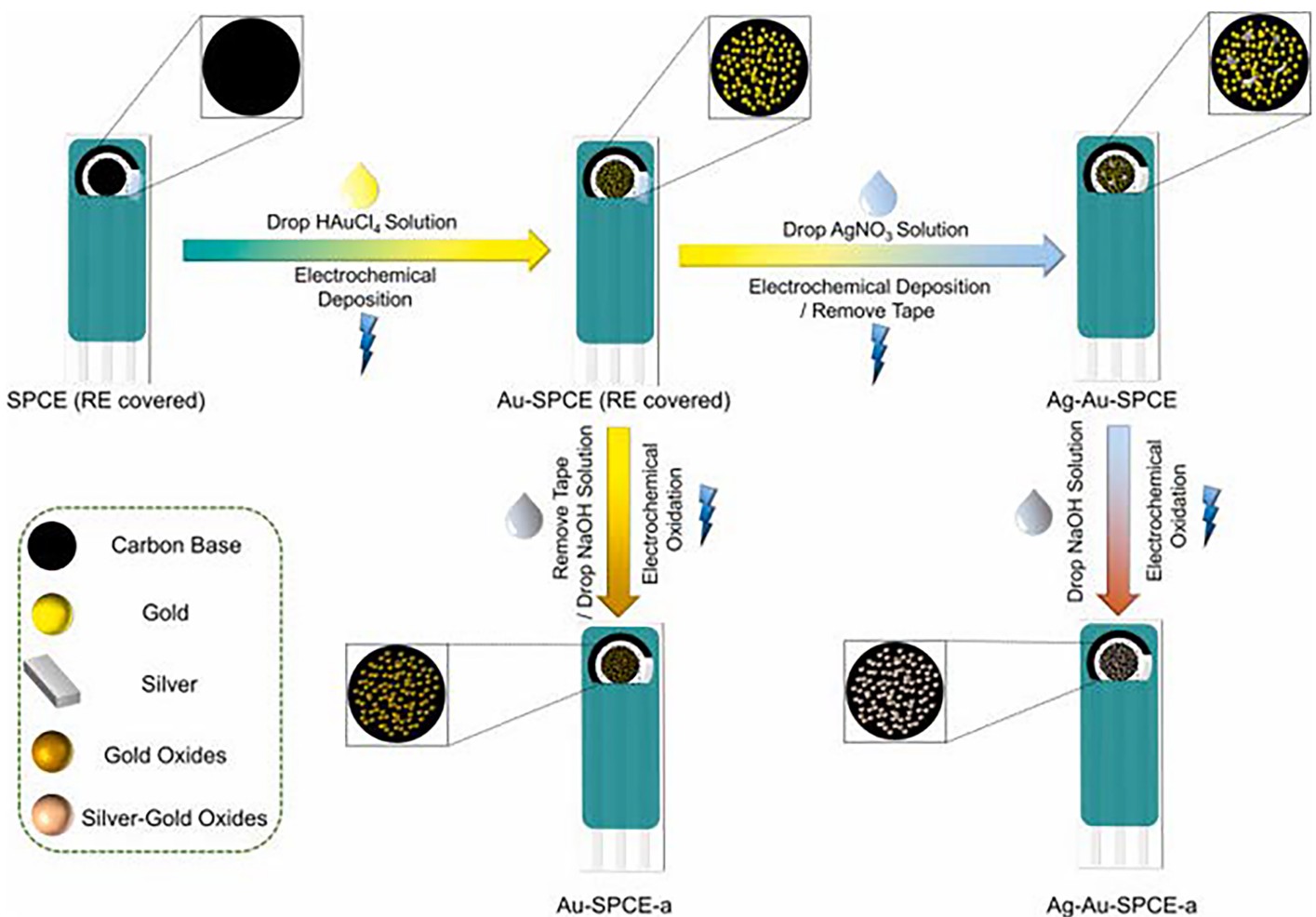

**Figure 2 Illustration of preparation of silver-gold bimetallic oxide nanoparticles.** Reprinted with permission from *Zhao et al. (2021)*. Copyright 2021 Elsevier.

artificial sweat. It was also able to monitor the levels of Cr (VI) and Cr (III) during the treatment of chromium-containing wastewater. With the help of a handheld dual-channel electrochemical device, it becomes effortless to simultaneously detect Cr (VI), Cr (III), and total chromium in different samples (*Zhao et al., 2021*).

A novel surface enhanced Raman scattering (SERS) sensor was developed for the first time to identify the increasing water and soil contamination caused by the excessive use of glyphosate, a widely used herbicide in agriculture. This sensor was created by capping reduced graphene oxide (rGO) with AgNPs on titanium dioxide ($TiO_2$) nanotubes. The sensor performed effectively in detecting methylene blue and glyphosate in water samples, with a limit of detection (LOD) of 10–14 M and 3 µg/L, respectively. Furthermore, the sensor demonstrated excellent reproducibility and repeatability, with a relative standard deviation of 2.0% and 4.0% respectively (*Butmee, Samphao & Tumcharern, 2022*). The development of these innovative sensors represents a notable progress in the field of environmental restoration endeavours. The key aspect is in its capability to

identify contaminants at extremely low concentrations. The sensor's exceptional sensitivity and dependability have the potential to enhance the monitoring and control of industrial contaminants, thereby making a significant contribution to the preservation of water and soil resources. These advances not only tackle an urgent environmental issue but also establish a new benchmark for identifying pollutants, opening possibilities for further study and technological advancements in the field of environmental science.

### AgNPs as antibacterial agents

Diversified AgNPs were synthesized biogenically from capsid structural proteins of bacteriophage (a naturally occurring bacterial virus) and applied both for the detection of heavy metal ions from water samples as well as for their antibacterial and anti-biofilm properties. The AgNPs were prepared by the bacteriophage mediated reduction of $AgNO_3$. NPs were found to have a particle size of 10 to 30 nm as confirmed by TEM. The developed NPs worked tremendously well and were found to successively inhibit the bacterial biofilm of *S. sciuri* and detect specifically $Cd^{2+}$ ions in water samples at a concentration of 100 μM (*Abdelsattar et al., 2022*). Biogenic recyclable core shell NPs consisting of FeO/AgNPs and FeO/Au NPs were synthesized using pomegranate fruit peel extraction. The formation of the dual types of NPs was confirmed by different spectral methods. In the UV-visible spectrum, absorbance peaks at 465 and 530 nm confirmed the formation of FeO/AgNPs and FeO/AuNPs. Through EMR analysis it was found that in the case of FeO/AgNPs, a 14 nm shell of AgNPs was found to surround the 13 nm Fe core whereas the average size of FeO/AuNPs were found to be less than 100 nm. The *in vitro* antimicrobial and antifungal abilities of the NPs were determined through zone of inhibition and mycelium inhibition methods. AgNPs are utilized in many medicinal applications for the prevention and treatment of fungal diseases. They are integrated into wound dressings, used as coatings for medical equipment, and used in antifungal lotions and ointments. AgNPs possess a wide range of antifungal properties, making them effective in treating infections caused by harmful fungus such as *Candida* and *Aspergillus* species (*Mussin & Giusiano, 2022*). The results indicated that the NPs possess a good range of antimicrobial properties against all types of microorganisms.

In a recent study by *Hidayat et al. (2022)*, synthesis of chitosan stabilized AgNPs immobilized to solid silica gel was developed in an eco-friendly and cost-effective manner (Figs. 3 and S1). MeOH was used to create the chi-AgNPs, which were then stabilized using white silica gel beads that had been coated with chitosan, also known as chi-SiG. This process was carried out under the influence of visible light. Using SEM, TEM, UV-visible, FTIR, and other techniques, it was demonstrated that the NPs had formed into a stable, solid, and dispersed form. The NPs are extremely stable because of the interaction between several functional groups on the surface of the chitosan and the $Ag^+$ ion. To combat the multidrug-resistant bacteria *S. aureus*, *E. coli*, and *B. subtilis* that were present in the air, the nanomaterial that had been developed was utilized as an air filter. The findings of the bactericidal study showed that the NPs had a strong inhibitory effect on the growth of bacterial cells in agar media. Additionally, they demonstrated a higher level of antibacterial
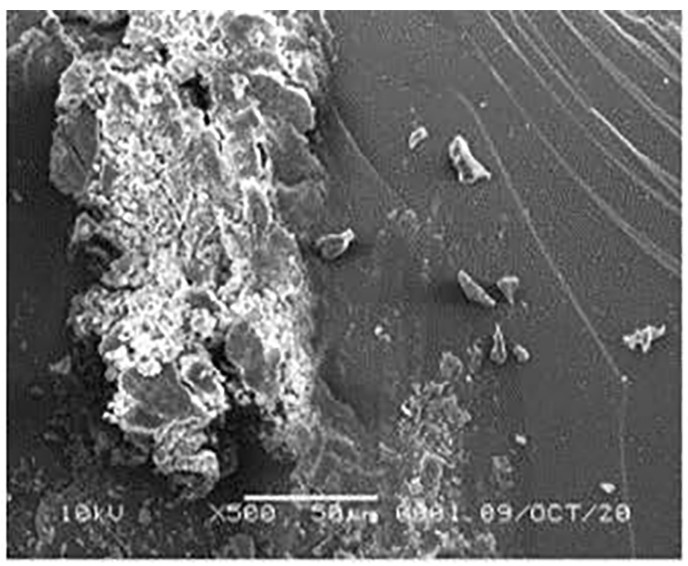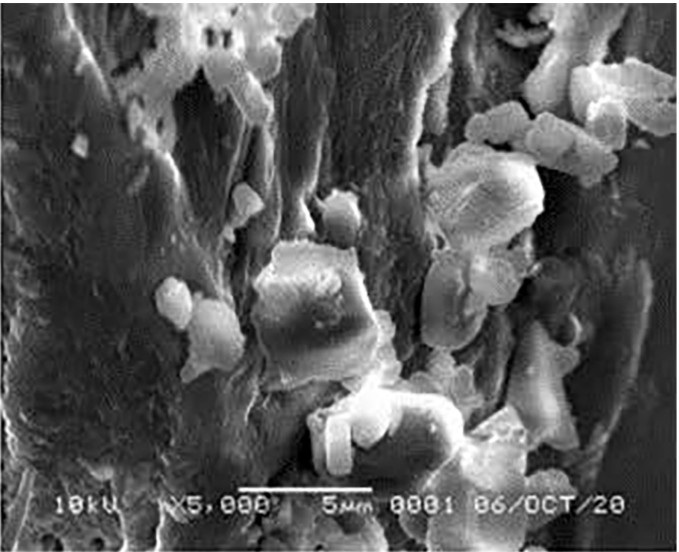

(a) SEM images of AgNPs-[chi-SiG].

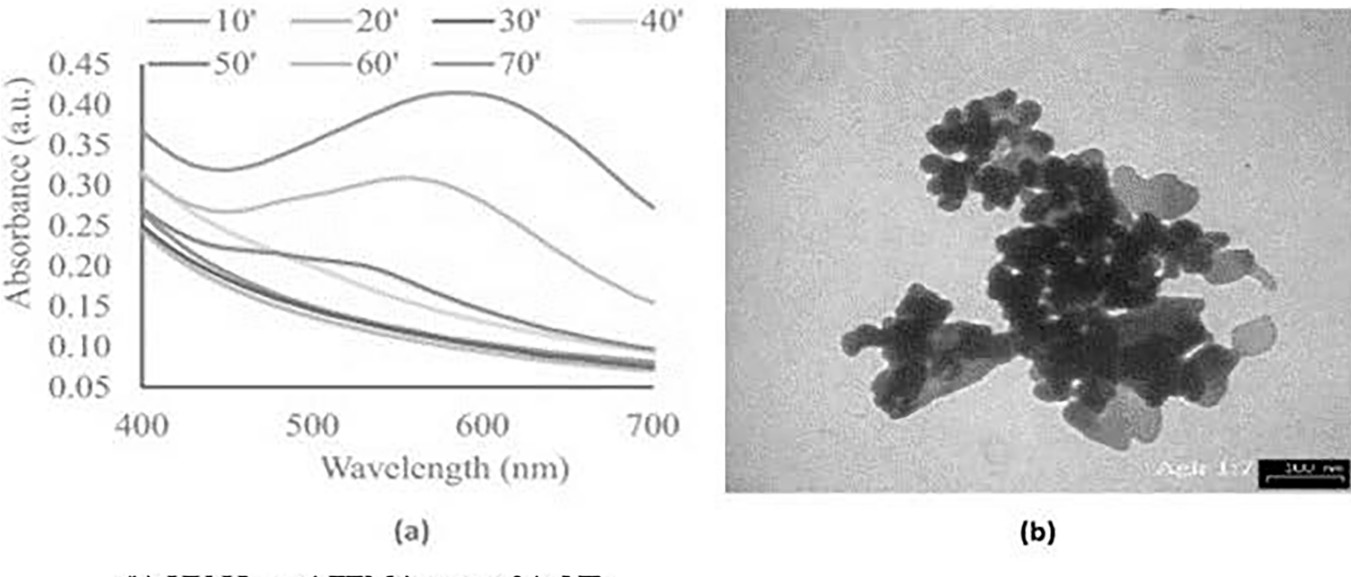

(b) UV-Vis and TEM image of AgNPs

**Figure 3** **(A and B) Characterization of AgNPs immobilized on chi-SIG.** Reprinted with permission from *Hidayat et al. (2022)*. Copyright 2022 Elsevier.                                                 

activity in the air against the *B. subtilis* bacterial strain, functioning as a filter for the air. Specifically, the AgNPs exhibited a mechanistic interaction with the proteins that were present on the bacterial cell wall as well as the phospholipids. This interaction led to the rupture of the cell wall, which in turn inhibited the development of the bacteria (*Hidayat et al., 2022*; *Zhang & Chen, 2009*).

In a recent study, a nanocomposite membrane containing AgNPs was synthesized and utilized for wastewater treatment (*Haider et al., 2016*). At first, the formation of aminated

polyether sulfone ($NH_2$-PES)—AgNPs occurred through the functionalization of $-NH_2$ on polyether sulfone. The AgNPs were immobilized on the $NH_2$-PES surface, resulting in the formation of the AgNPs-APES. The nanocomposite membrane was characterized by TEM, FTIR, SEM, EDAX, XRD, and other techniques to establish synthesis and immobilization. The findings from the experiments revealed that NPs with diameters ranging from 5 to 40 nm were generated and immobilized on the surface of the APES. The NPs have been evaluated for their capacity for inhibiting microorganisms in wastewater. The study revealed that the nanocomposite membrane made with amino functionalized AgNPs-APES exhibited greater antibacterial effectiveness compared to unfunctionalized AgNPs-PES. This could be attributed to the extended lifespan of the membrane, lasting approximately 25 days, because of enhanced and regulated release of $Ag^+$ ions. The NPs function by impeding the biofouling capability of the pathogens, namely by accumulating AgNPs on the cell wall of the microorganisms, resulting in the rupture of their cell wall and finally preventing bacterial proliferation (Fig. S2).

Recent observations have shown that silver-based porous nanocomposites (AgNCs) have effectively killed bacteria and viruses in drinking water, achieving a reduction of 99% to 100% (*Bhardwaj et al., 2021*). Multiple studies have demonstrated that AgNPs have the capability to eliminate around 99.99% of *E. coli* bacteria and MS2 bacteriophage viruses (*Bhardwaj et al., 2021*). Nanocomposites made of multi-walled carbon nanotubes were produced with FeO and AgNPs for the treatment of wastewater that was contaminated with bacteria. The TEM, SEM, XRD, XRF, and EDAX techniques were utilized to validate the molecular makeup, crystal structure, material, shape, and surface properties of the NPs. The NPs exhibited a noteworthy antibacterial capability in relation to *E. coli*, with a minimum bactericidal concentration of 200 µg/ml. The bacterial growth inhibition time was set at 8 h (*Ali et al., 2017*).

Recently, *Konduri et al. (2024)* reported on the green manufacturing of AgNPs from an aqueous extract of the leaves of *Hibiscus tiliaceus* L., as well as its application in the degradation of dyes, antioxidant activity, antibacterial activity, and anticancer activity. X-ray crystallography (XRD) confirmed that the AgNPs were in crystal form, and analysis using Fourier transform infrared (FT-IR) spectroscopy revealed that plant metabolite functional groups had a role in the reduction and stability of AgNPs. The investigations that were conducted using UV–vis spectroscopy, dynamic light scattering (DLS), and zeta potential showed that the AgNPs were produced in colloidal form with an average size of 88.10 nm and were stable (−49 mV). Both the field emission scanning electron microscopy (FE-SEM) and the high-resolution transmission electron microscopy (HR-TEM) techniques were able to confirm that the AgNPs were spherical in shape and had a particle size that ranged from 30 to 35 nanometers. Based on the results of total antioxidant, DPPH, and reducing power experiments, the AgNPs demonstrated the potential to exhibit antioxidant activity. Using the zone of inhibition assay, the biosynthesized AgNPs demonstrated a broad spectrum of antibacterial activity against Gram-negative and Gram-positive bacteria. There was a significant anticancer activity demonstrated by AgNPs on MCF-7 cells, with an IC50 value of 65.83 µg/mL or higher. According to the

**Table 2  AgNPs for dye removal in wastewater treatment.**

| No | AgNPs-composites | Type of pollutant | Treatment efficiency | References |
|---|---|---|---|---|
| 1 | TiO$_2$/CNTs/AgNPs/Surfactant nanocomposite | Methylene blue (MB) dye | Degraded in 180 min; 0.5 g$^{L-1}$, 100% | *Azzam et al. (2019)* |
| 2 | CNF/PEI/Ag NPs composite | MB | 96% after 4 min | *Zhang et al. (2020)* |
| 3 | CAg-NPs | Congo red (CR), MB, malachite green (MG) | MB: 93.29; MG: 83.73;4-NP: 88.9 | *Elbakry et al. (2022)* |
| 4 | rGO-AgNP (graphene oxide silver nanoparticle hybrid nanocomposite) | Direct blue-14 | 95.41% | *Choudhary et al. (2021)* |
| 5 | GO–ZnO–Ag | MB | 100%, 40 min | *Naseem et al. (2020)* |
| 6 | AgNPs/holocellulose nanofibrils (AgNPs/HCNF) | MB | 94–98%, catalytic activity with five cycles | *Bandi et al. (2020)* |
| 7 | AgNPs/ZIF-8 composite | MB and CR | MB: 97.25%; CR: 100% | *Chandra & Nath (2020)* |
| 8 | AgNPs impregnated sub-micrometer recrystalline jute cellulose (SCJC) particles | CR and MB | 100%, 14 min with 0.005 mg/mL | *Rabbi et al. (2020)* |
| 9 | AgNPs | Reactive green 19A, R blue 59, R red 120, R red 141, and R red 2 | 180 min, 50; 35% fourth and fifth cycles | *Saratale et al. (2020)* |
| 10 | Ag@MGO-TA/Fe$^{3+}$ nanocomposite | MB | 0.05 mg/mL | *Lai et al. (2022)* |
| 11 | CH-AgNPs | Orange and blue dyes | 97.4 and 100% | *Gola et al. (2021)* |
| 12 | MMT/Ag nanocomposite | MB | 99.90% for 25; 96.50% for 50; 89% for 100 and 81.14% for 200 ppm | *Liao et al. (2018)* |
| 13 | Ag/rGO nanocomposite and Ag/rGO/CA/TFC membranes | MB | 98%; 92% | *Vatanpour et al. (2022)* |
| 14 | AgNPs decorated on nanostructured porous silicon | MB | Degradation rate 8.6/min | *Naveas et al. (2022)* |
| 15 | BaTiO$_3$/AgNPs | MB and ciprofloxacin | 72 and 98% | *Masekela et al. (2022)* |

findings of the study, green produced AgNPs have the potential to be extremely relevant in the field of biomedicine as antioxidant, antibacterial, and anticancer agents (Fig. S3).

### AgNPs in dye degradation

AgNPs have been thoroughly investigated for their ability to break down synthetic dyes, which pose a significant environmental threat because of their widespread use in many industries and subsequent release into water bodies (*Mehta et al., 2021*). Recent studies have emphasized the function of biosynthesized AgNPs in removing colorants from industrial wastewater (Table 2). An example of a successful process is the use of silver-manganese oxide NPs to degrade Malachite Green dye using photocatalysis. This method has demonstrated high efficiency in breaking down the dye when exposed to sunshine (*Pal et al., 2013*). The main benefit of utilizing AgNPs for dye degradation lies in their exceptional efficacy in eliminating harmful dyes from water, hence reducing the environmental impact. The eco-friendly green synthesis of AgNPs is conducted using mild conditions, making it a sustainable approach. Moreover, the utilization of AgNPs in the process of dye degradation can result in substantial enhancements in the quality and safety of water (*Palani et al., 2023*). Additionally, the AgNPs demonstrated their potential as
catalysts when combined with sodium borohydride (NaBH4), a reducing agent, to facilitate the degradation of methylene blue (MB), methylene orange (MO), and methylene green (MG) dyes. The degradation efficiency of the AgNP catalyst in the presence of NaBH4 for 15 min was found to be 12.8%, 26.92%, and 47.56% for MO, MB, and MG, respectively. The basic mechanism of degradation involves the use of AgNPs as an electron relay, which triggers the transfer of electrons from the BH4- ion (donor $B_2H4/BH_4$) to the acceptor (acceptor LMB/MB), resulting in the reduction of the dye. The $BH4^-$ ion is adsorbed on the surface of NPs, leading to electron transfer from the $BH4^-$ ion to the dye through the NPs (*Fairuzi et al., 2018*). AgNPs provide a sustainable and effective method for addressing dye contamination. Nanotechnology has proven to be beneficial in the breakdown of dyes. Different types of metallic and non-metallic NPs can eliminate a wide range of contaminants from wastewater. Moreover, most of the literature focused on using a single dye, with only a limited number of studies exploring the use of NPs for multiple dyes in combination. Hence, this study serves as a valuable reference for researchers to recognize the potential of various NPs in treating different synthetic dyes, both individually and in combination, for enhanced efficacy. In addition, it is crucial to investigate the effective degrading capabilities of catalysts for large-scale industrial applications. Furthermore, it is essential to examine how catalysts can be properly integrated with existing technologies, ensuring their stability and recyclability in real-world scenarios for long-term sustainability. In addition, there are certain challenges linked to the utilization of AgNPs. An important drawback is the potential toxicity of NPs, which can prove hazardous to human health and the environment if not adequately controlled. Furthermore, the persistent stability and potential buildup of AgNPs in the environment are still subjects of continuing investigation. Ultimately, although AgNPs show potential as a viable method for dye degradation, it is imperative to carefully consider and manage the associated advantages and hazards. Continual research and development are crucial for maximizing their effectiveness and safety in environmental remediation.

### AgNPs in water treatment

Water is an essential requirement for living, yet obtaining clean and safe drinking water continues to be a difficult task in numerous regions across the globe. Microorganisms, including bacteria and viruses, as well as organic contaminants, provide substantial hazards to human well-being. Conventional water treatment systems, although successful, have restrictions, specifically in terms of effectiveness, expense, and the capacity to eliminate specific types of pollutants. Advancements in nanotechnology have recently brought up new and creative solutions, with AgNPs showing great potential in water treatment. A prominent application of silver NPs in water treatment is their utilization as antibacterial agents. AgNPs demonstrate a wide-ranging ability to combat various types of pathogens, such as bacteria, viruses, and fungi, due to their broad-spectrum antibacterial activity. The main reason for this is their capacity to emit silver ions ($Ag^+$), which interact with the membranes and internal components of microorganisms, resulting in the death of the cells. NPs have a greater surface area to volume ratio, which improves their interaction

with microbes, resulting in higher effectiveness compared to larger quantities of silver (*Li et al., 2008*; *Yin et al., 2013*; *Panda, Chakraborty & Krishna, 2023*).

AgNPs can be integrated into many water treatment systems, including filtering membranes and coatings for pipes and storage tanks. One example is the utilization of silver-impregnated activated carbon filters to eliminate microbiological impurities from drinking water. The integration of physical filtration and the antibacterial property of silver guarantees a heightened degree of water purification. Moreover, AgNPs have the potential to be utilized in portable water purification equipment, hence rendering them highly important for emergency relief operations and in regions where centralized water treatment facilities are not readily available.

In addition to their antibacterial characteristics, AgNPs also contribute to the elimination of chemical pollutants from water. AgNPs can be modified to have particular surface properties that allow them to absorb and break down different types of organic contaminants, such as insecticides, medicines, and colorants. This is accomplished by mechanisms such as photocatalysis, in which AgNPs serve as catalysts when exposed to light, causing the breakdown of complex chemical compounds into less damaging ones. Furthermore, AgNPs have been researched for their capacity to eliminate heavy metals from polluted water. These substances have the ability to create compounds with metal ions including lead, mercury, and cadmium, which helps in their elimination by either adsorption or precipitation. This application is especially important in the treatment of industrial wastewater, where the presence of heavy metals is a considerable issue. The use of AgNPs with conventional water treatment technologies significantly augments their efficacy. By integrating AgNPs with membrane filtering systems, both the elimination of microorganisms and the occurrence of membrane fouling are enhanced, resulting in a prolonged lifespan for the membranes. AgNPs can be combined with other nanomaterials, like titanium dioxide ($TiO_2$) or graphene oxide, to produce composite materials that have several functions and exhibit synergistic features. These materials are particularly useful for advanced applications in water treatment (*Li et al., 2008*; *Yin et al., 2013*).

There have been many studies that focused on the synthesis of AgNPs and their utilization in water treatment, primarily for eliminating three contaminants, namely pesticides, heavy metals, and microbes. Several synthesis strategies have been documented for the production and analysis of AgNPs and water treatment with silver has been conducted to exploit their antibacterial properties. which is largely due to three primary modes of action, name modification of membrane characteristics, impairment of DNA/RNA and/or proteins, or liberation of Ag (I) within the cell cytoplasm (*Sartori et al., 2023*).

## Cost-estimation for AgNPs in relation to other NPs used in environmental remediation

Depending on the exact application, type of nanoparticle, and method of production, the cost of environmental remediation NPs might vary greatly. In this article, we present a synopsis of the cost calculation for AgNPs (Table S1) compared to other NPs utilized in environmental remediation, including $TiO_2$, $Fe_3O_4$, and ZnO. Although they are more costly to make, AgNPs are highly recognized for their catalytic powers and antibacterial

characteristics. The price of silver, the method of synthesis, and the size of manufacturing are three of the many factors that affect the cost of AgNPs. Many other ways can be employed to produce AgNPs, including physical processes, chemical reduction, and green synthesis. Although green synthesis methods using biological materials or plant extracts can be environmentally friendly, they may be more expensive than traditional nanoparticle manufacturing methods. Based on factors such as purity, particle size, and the chosen synthesis process, the price of AgNPs can differ greatly, usually falling between $100 and $500 per gram. AgNPs are far more costly than those of titanium dioxide, iron oxide, and zinc oxide, which are more often utilized in environmental remediation. Factors contributing to the high price of AgNPs include the complexity of synthesis methods, the volume of manufacturing, and the high price of silver. The distinctive benefits of AgNPs, especially their antibacterial characteristics, are not often matched by their widespread use due to their high cost. To find a happy medium between efficacy and economic feasibility, other NPs, such as $TiO_2$ and ZnO, offer more affordable alternatives for certain remediation tasks (*Iravani et al., 2014*; *Thakkar, Mhatre & Parikh, 2010*; *Islam, Jacob & Antunes, 2021*).

## Advantages and drawbacks of AgNPs

The distinctive characteristics and diverse uses of AgNPs have attracted a lot of interest, especially in the fields of antibacterial action and environmental remediation (Table S2). Viruses, bacteria, and fungi are only some of the microbes that are severely inhibited by AgNPs' potent antimicrobial properties. A key mode of action is their capacity to release silver ions (Ag+), which damage internal components and cell membranes of microbes. The low concentration required to achieve AgNPs' high efficacy makes them a promising material for use in water and air purification systems, coatings, and medical devices. The amount of material needed for effective treatment can be minimized thanks to its efficiency. The antimicrobial capabilities of AgNPs can be applied to a diverse array of products through their incorporation into textiles, polymers, and coatings. They find application in consumer goods, air and water filtration systems, wound dressings, and medical devices. However, ecosystems and human health could be jeopardized by releasing AgNPs into the environment. Bioaccumulation and biomagnification are possible outcomes of their buildup, which can be harmful to aquatic life. Misuse and overuse of AgNPs also has the potential to cause the rise of bacteria and other microbes that are resistant to antibiotics. The efficacy of AgNPs may thus be diminished because of this possible resistance (*Mo, Zhou & He, 2022a*; *Nakamura et al., 2019*).

Producing AgNPs is becoming cheaper due to advances in nanotechnology, however when compared to more traditional materials and procedures, nanoparticle production costs are still high. Their high price tag thus might prevent NPs from being widely used in some contexts. To avoid polluting the environment, it is also essential to handle and dispose of AgNPs safely. NPs, which can cause health and ecological problems, can be

released into the environment as a result of improper disposal practices. With their antibacterial activity, efficiency, and adaptability, AgNPs provide a plethora of benefits. Nevertheless, there are a number of obstacles to overcome, including the fact that they may be poisonous, have an effect on the environment, pose a threat of microbial resistance, be economically unviable, and require proper disposal and handling. For the appropriate and long-term usage of AgNPs in different applications, it is crucial to weigh these benefits and downsides (*Ngcongco, Krishna & Pillay, 2023*).

## CONCLUSIONS

This study concludes by emphasizing the significance of managing environmental contaminants with AgNPs. Environmental pollution is increasing globally each day due to various types of manmade activities which results in multiple kinds of air, water, and soil borne diseases. Nanomaterials made up of AgNPs have found a great deal of environmental applications because of their safe and effective synthesis, tiny particle size with high surface area as well as biocompatibility. At low concentration they are found to be more harmful towards the pollutants including dyes, micro-organisms, pathogens, and heavy metal ions, having no adverse effect towards the healthy cells including soil micro-organisms. In this review the authors briefly elaborate the versatile synthesis and applications of AgNPs in the form of photocatalyst, electrochemical sensor, fluorogenic sensor *etc.* towards the detection of hazardous pollutants in the air, water, and soil medium. All the synthesized nanomaterials possess low LOD value with high sensitivity and selectivity towards the detection of pollutants. Due to their biogenic nature, it can also be further recycled and reused. This review will help the future researchers towards the more economical and efficient innovation of AgNPs carrying nanoprobes which can be efficient to detect multiple groups of pollutants at the same time with low LOD value along with good reproducibility in environment remediation.

### Funding
This work was funded by the Deanship of Graduate Studies and Scientific Research at Jouf University under Grant No. (DGSSR-2023-01-02004). The funders had no role in study design, data collection and analysis, decision to publish, or preparation of the manuscript.

### Grant Disclosures
The following grant information was disclosed by the authors:
Deanship of Graduate Studies and Scientific Research at Jouf University: DGSSR-2023-01-02004.

### Competing Interests
The authors declare that they have no competing interests.

## Author Contributions

- Suresh Babu Naidu Krishna conceived and designed the experiments, performed the experiments, analyzed the data, prepared figures and/or tables, authored or reviewed drafts of the article, and approved the final draft.
- Abdul Gaffar Sheik conceived and designed the experiments, performed the experiments, analyzed the data, prepared figures and/or tables, and approved the final draft.
- Karen Pillay conceived and designed the experiments, performed the experiments, analyzed the data, authored or reviewed drafts of the article, and approved the final draft.
- Manhal Ahmed Hamza conceived and designed the experiments, analyzed the data, authored or reviewed drafts of the article, and approved the final draft.
- Mohammed Yagoub Mohammed Elamir performed the experiments, analyzed the data, prepared figures and/or tables, authored or reviewed drafts of the article, and approved the final draft.
- Samy Selim conceived and designed the experiments, analyzed the data, prepared figures and/or tables, authored or reviewed drafts of the article, and approved the final draft.

## Data Availability

This is a literature review.

## Supplemental Information

Supplemental information for this article can be found online at http://dx.doi.org/10.7717/peerj.18191#supplemental-information.

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
