# Peer review of "Nanotechnology in action: silver nanoparticles for improved eco-friendly remediation"

_PeerJ, doi:10.7717/peerj.18191_

## Round 0.1 · original submission · Major Revisions

I agree with the four reviewers, major revisions. Authors should reply to all reviewers' comments

Reviewer 1 ·

Basic reporting

Is the review of broad and cross-disciplinary interest and within the scope of the journal?
#Yes
Has the field been reviewed recently? If so, is there a good reason for this review (different point of view, accessible to a different audience, etc.)?
#Yes
Does the Introduction adequately introduce the subject and make it clear who the audience is/what the motivation is?
#Yes

Experimental design

Is the Survey Methodology consistent with a comprehensive, unbiased coverage of the subject? If not, what is missing?
# Yes
Are sources adequately cited? Quoted or paraphrased as appropriate?
# Yes
Is the review organized logically into coherent paragraphs/subsections?
#Yes

Validity of the findings

Is there a well developed and supported argument that meets the goals set out in the Introduction?
# Yes
Does the Conclusion identify unresolved questions / gaps / future directions?
# No..

Additional comments

The review paper aims to provides a review of the applications of silver nanoparticles (AgNPs) in environmental remediation. The authors first discussed the various methods of synthesizing AgNPs, including physical, chemical, and biological approaches. The authors discussed the emergence of nanoscience and the potential of AgNPs in addressing environmental challenges. The paper then highlights the unique applications of of AgNPs in environmental remediation, such as their antibacterial, dye degradation & sensing probes for various types of pollutants. Although the review is organized well & provides a good coverage of some of the applications of Ag NPs covered under the theme. However the paper needs to be improved further as most topics covered under environmental remediation has not been discussed properly. Further, I recommend the paper to be supplemented with the following sections.
1. Applications in Water Treatment
Use of silver nanoparticles in removing heavy metals, organic pollutants, and pathogens from water.
Case studies and examples
2. Applications in Soil and Air Remediation
Role of silver nanoparticles in soil remediation, including removal of heavy metals and organic pollutants. Air purification applications, focusing on the degradation of airborne pollutants.
3. Mechanisms of Action in Environmental Remediation
How silver nanoparticles interact with contaminants.
Mechanisms of pollutant degradation, adsorption, and catalysis.

Annotated reviews are not available for download in order to protect the identity of reviewers who chose to remain anonymous.

·

Basic reporting

The article titled - "Nanotechnology in action: silver nanoparticles for improved environmental remediation" talks about the latest developments in the field of AgNPs as an environmental remediation agent. The use of English has been excellent with few modifications required. The references are upto-date which confirms a thorough literature review has been conducted by the authors. The introduction does efficiently highlights the current state of problem. The manuscript as a whole can benefit audiences or readers from multidisciplinary fields. However I have several concerns on the authors selecting this kind of study as a review. There have been multiple numbers of reviews on AgNPs and the authors have failed to state how their article is novel then the others. The article although discusses several research studies but fails to make an impact as far as different sections are concerned.
I am sorry to say that the article lacks in-depth analysis and proper presentation to make it stand well against other articles on the same topic. The authors actually have missed to present their novelty in the quest to discuss multiple research done till date in the field.
1. Please add a statement of novelty to the abstract as well as the introduction part of the manuscript.
2. The abstract is missing with a a concluding remark.
3. Under Biosynthesis of silver nanoparticles, how do the authors classify the biological synthesis or green synthesis of AgNPs? Is it a Top-Down or Bottom-Up approach? please justify.

Experimental design

Although the review is a commendable work but lacks on the following grounds -
4. Although the methodologies have been excellently framed, but the organization of the review is abrupt.
5. The review only talks about contaminated soils and water. What about the role of AgNPs in remediation of polluted air ?
6. Line 218-219: Provide an account of research done on both these aspects in form of a table .. Include the organism, Size of the AgNPs, Conditions like Temp., pH, any other, Approach (Bioreduction or Biosorption), and latest references supporting it.
7. Under "Silver nanoparticles as sensors": How do silver nanoparticles act as a good sensor ? Is there any particular mechanism?
8. Table 1 is not referred anywhere within the text. Please refer at the appropriate place.
9. Under AgNPs as antibacterial agents: AgNPs have been found to pose as an excellent antibacterial agent. The authors have quite creatively discussed different research projects. However, the section starts abruptly with examples. Please provide a generalized section along with the mechanism of AgNPs against harmful bacterial consortium.
10. Under AgNPs in dye degradation: Please formulate and describe an overall mechanism for the same.
11. Line 438-440: What is the real mechanism behind this efficacy ?
12. Before the section - "Future Perspectives" Before this section the authors must add a section depicting advantages and drawbacks of the AgNPs over other nanoparticles. Future perspectives should be based on delineating the drawbacks.

Validity of the findings

13. It has been found that authors suggest the AgNPs to be more cost-effective and environmental friendly. It is required by the author to provide an overall cost-estimation (in tabular form along with proper discussion) for AgNPs in relation to other nanoparticles used in environmental remediation. This will be a very interesting part of the manuscript and can draw several citations.

Reviewer 3 ·

Basic reporting

1. There are more grammatical and typing errors in the text. Please check the manuscript thoroughly to fix them. You can use different software Add-ons to correct them.
2. The title of the manuscript can be modified little bit related to the content.
3. The authors should add more technical information regarding the silver nanoparticles and if possible can add antifungal work.
4. The authors must specify the characterization equipment used (brand, model) in the methods section.

Experimental design

5. The purpose of the study undertaken, what are you trying to solve?
6. Authors must extensively proofread the article for publication.
7. Important conclusions based on the obtained results Potential applications. Therefore, it is suggested that the Abstract and conclusion be modified as per the suggestions given above and below. Please start the abstract by a short introduction of the current problem(s) and the solution, based on the current study, in one or two lines.
8. The abstract part needs to rewrite in a way to define the exact novelty and originality of your work.

Validity of the findings

9. The introduction must be completed by clarifying the main objectives of the research and by motivating the experimental strategy adopted by authors.
10. In introduction section, author should be clarify the advantages of synthesis nanoparticles and give more knowledge about their application
11. Please specify the impact of your study in the introduction.
12. Add all materials used in Materials section and add references to all the methods.
13. Replace Figure 6 with a better one.
14. Photocatalytic degradation of silver nanoparticles is unknown and unclear. What is the effect of nanoparticle concentration, pH, and temperature on the photocatalytic process?
15. Please explain how organisms were inoculated into one petri dish, and then the antibacterial
activity of nanoparticles was determined for each organism in this petri dish. Concentration
of FeO nanoparticles?

Reviewer 4 ·

Basic reporting

Basic reporting is good but must be improved before the publication.

Experimental design

This is review article

Validity of the findings

Good

Additional comments

Peer J- Nanotechnology in action: silver nanoparticles for improved environmental remediation (#100296)
Thanks for giving me the opportunity to review the article entitled "Nanotechnology in action: silver nanoparticles for improved environmental remediation" I have reviewed the article thoroughly and recommend this manuscript for major revision. I hope that after considering the below suggestions, the manuscript should be understandable and reflect the true sense of outcomes. The suggestions are given below:
1. The critical component of the abstract namely concise problem statement with novelty; method; results discussed, conclusion with possible application need to be explicitly highlighted and revised. Abstract should be more numerical.
2. In the introduction part, please provide a critical literature review to describe the research gaps of previous works.
3. The language in this manuscript needs to be improved. There were many grammar mistakes, such as the wrong tense and a set phrase.
4. Please give a frank account of the strengths and weaknesses of the article. Include specific, detailed comments regarding the originality, scientific quality, relevance to the field of this journal, and presentation. Check the need for tables and figures, and the adequacy of the references.
5. The discussion part should be better organized and more concise. The discussion part is too long and is filled with too much information without being properly organized. Your findings and other people’s finding are mixed together. It would be better if you first state about your major findings, then list the literatures that support your findings or not support your findings (this section should be as concise as possible), and finally discuss about the reason why your findings is different from the previous (or what makes your study original compared with other study).
6. The conclusion should highlight the key finding from the content of this manuscript. Conclusions should highlight the insights and the applicability of your findings/results for further work. Likely future study needs to be integrated to the conclusion.
7. Author should be clearly indicated that how is the benefited to human society in general life of this research directly or indirectly. Besides, it would be great if the authors add more information about the drawbacks of the utilized methods if any.
8. The author should give the full names of the abbreviations at the first appearance after the conclusion with a table form.
9. Several typesetting errors are present throughout the text and stylistic features should be homogenized as well.
10. To enhance the manuscript's competitiveness for potential publication, I recommend incorporating more innovative elements into your research. This could involve introducing new data sets, developing novel models, or exploring more profound empirical investigations. Strengthening the originality and depth of your research will significantly improve its chances of acceptance.

---

## Round 0.2 · Minor Revisions

Dear authors,

I have taken over the handling of your manuscript as an Academic Editor. While I consider the manuscript overall well written some aspects still need work and revision before the manuscript can be considered for publication:

1. The methodology description needs to be specific to the review paper, detailing the selection criteria of included literature, keywords used, the time scale of included papers, etc.
2. In Figure 1, the synthesis methods of nanoparticles have been divided into “toxic” and “non-toxic”. The rationale for such categorization is unclear because often the Ag NPs resulting from “green synthesis” are more efficient in antibacterial applications (i.e., more toxic) due to containing bioactive compounds from the reaction mixture (plant, bacterial or fungal extracts). The scheme should be modified so that labels of “toxic” and “nontoxic” are removed because these are misleading and unclear. Also, it should be indicated that the listed methods apply for Ag NP synthesis and not all NPs because these methods do not apply to all types of nanoparticles.
3. In the section “AgNPs in Soil and Air Remediation”, starting in line 512, the paragraph about soil remediation does not contain any information about Ag NPs, but discusses other types of NPs. If there are no studies about Ag NPs in soil remediation to discuss, the section about soil remediation should not be included in the review. The current text in lines 513-538 has no relevance to the review about Ag NPs and should be removed from the manuscript.
4. The sections “Future perspectives” and “Conclusions” contain mostly identical text. These sections should be rewritten.
5. AgNPs have been defined in the Introduction, line 86. There is no need to define them again in lines 296, 391, 396, 405, 409, 415, 428, 466, 468, 476, 499, 548, 553, 585, 632 nor nanoparticles again in lines 340 and 345.

Reviewer 1 ·

Basic reporting

No comments

Experimental design

No comments

Validity of the findings

No comments

Additional comments

No comments

·

Basic reporting

No Comment

Experimental design

No Comment

Validity of the findings

No Comment

Additional comments

Some revisions are still required
Still, the article needs to have a thorough grammar check. Please check every sentence bit by bit for formatting or grammatical errors.
The first page of AgNPs in water treatment has no reference. Please add them.
The section AgNPs in soil and water remediation needs more inclusion of references.
The section related to cost estimation also has no references to support the author's statements.
The "Advantages and Drawbacks" section also lacks any references.

Reviewer 4 ·

Basic reporting

The questions raised were addressed by the authors. The revisions are accepted.

Experimental design

Correct

Validity of the findings

Correct

Additional comments

The questions raised were addressed by the authors. The revisions are accepted.

---

## Round 0.3 · Minor Revisions

Dear authors,

While some changes have been made to the manuscript, the following requested modifications have not been incorporated in the revised manuscript despite of brief remarks in the rebuttal letter stating that these have been “corrected”. Please explain in your response letter, which changes are made and where in the manuscript.

Please address the following aspects in the manuscript:
1. In the Methodology section, please list which selection criteria were used for inclusion of literature in the review, which keywords were used to search the literature, what was the time scale of included papers, etc.

2. In the section “AgNPs in Soil and Air Remediation”, starting in line 505, the paragraph about soil remediation does not contain any information about Ag NPs, but discusses other types of NPs. The section about soil remediation should not be included in the review if it does not contain information about Ag NPs.

3. The sections “Future Perspectives” and “Conclusions” are not identically worded in the new version, however, the contents of the two sections are still identical. In the section “Future Perspectives” no future perspectives are discussed. Please modify or delete the section.

4. Please check the whole manuscript and use “NP” instead of “nanoparticles (NPs)” or “nanoparticles” and instead of “silver nanoparticles” use “AgNPs” throughout the text, to be consistent with using acronyms. Also, “engineered nanoparticles (ENPs)” is repeatedly defined in the manuscript. After explaining the abbreviation, it should be used consistently.

---

## Round 0.4 · accepted · Accept

Dear authors,

Thank you for addressing all the comments. The manuscript is now ready for publication.